# Improving Online Source-Free Domain Adaptation for Object Detection by Unsupervised Data Acquisition

Xiangyu Shi[1], Yanyuan Qiao[1], Qi Wu[1], Lingqiao Liu[1], and Feras Dayoub[1]

Australian Institute for Machine Learning, The University of Adelaide
{xiangyu.shi, yanyuan.qiao, qi.wu01, lingqiao.liu,
feras.dayoub}@adelaide.edu.au

**Abstract.** Effective object detection in autonomous vehicles is challenged by deployment in diverse and unfamiliar environments. Online Source-Free Domain Adaptation (O-SFDA) offers model adaptation using a stream of unlabeled data from a target domain in an online manner. However, not all captured frames contain information beneficial for adaptation, especially in the presence of redundant data and class imbalance issues. This paper introduces a novel approach to enhance O-SFDA for adaptive object detection through unsupervised data acquisition. Our methodology prioritizes the most informative unlabeled frames for inclusion in the online training process. Empirical evaluation on a real-world dataset reveals that our method outperforms existing state-of-the-art O-SFDA techniques, demonstrating the viability of unsupervised data acquisition for improving the adaptive object detector.

**Keywords:** Domain Adaptation · Visual Perception · Object Detection

## 1   Introduction

Autonomous vehicles operating in diverse and constantly changing environments require robust object detection systems. However, the performance of trained detectors significantly drops when they encounter unfamiliar scenarios. The real world is rarely predictable, with environments varying widely in their characteristics, leading to domain shifts that can severely compromise the effectiveness of pre-trained object detection models. This is a challenging task and many existing approaches focus on addressing this issue [1, 3, 15, 18, 23, 29].

The challenge lies in equipping vehicles with the ability to adapt to new domains "on-the-fly", without the need for manually labelling new data or undergoing an exhaustive re-training process. Online source-free domain adaptation (O-SFDA) [20, 23, 27] offers a promising avenue to address this challenge by transferring knowledge from a source domain to a target domain in an online manner. This means that the adaptation occurs concurrently with deployment in the target domain, contrasting with traditional methods that may require source

data batches or offline processing [8, 14, 24, 28]. Current O-SFDA methods for object detection, like those described by [23], utilize a transformer-style memory bank to store and use information from observed frames for future adaptation. However, these methods face limitations due to high computational costs and overreliance on redundant data. Adapting using every frame can result in severe class imbalances, such as more frequent detection of common objects like cars and people, and less frequent detection of rare objects like motorbikes, leading to poorer detection performance for rare classes.

To address these issues, we employ an unsupervised data acquisition strategy based on incremental online clustering to dynamically identify and integrate both the most informative unlabeled frames and those containing rare categories. We augment this strategy by optimizing a dual architecture consisting of a teacher model and a student model based on the Mean Teacher framework [21]. Our method aims to reduce computational resource usage while enhancing object detection adaptivity in previously unencountered deployment environments.

We conducted extensive experiments across four datasets, Cityscapes [2], Sim10k [6], Cityscapes-Foggy [16], and SHIFT [19] with three domain adaptation processes that include real-world scenarios with varying complexities. Our results demonstrate that our method consistently outperforms existing state-of-the-art O-SFDA techniques in object detection [23] for 2.3 mAP under Cityscapes to Cityscapes-Foggy, 9.0 mAP under Sim10k to Cityscapes, and 9.3 mAP under SHIFT to Cityscapes.

In summary, we propose a novel approach to improve O-SFDA for object detection through an unsupervised data acquisition methodology based on incremental online clustering. This data acquisition strategy selects the most informative and category-rare frames for online training. To solve the overlapping frame and class imbalance issues, our proposed data acquisition contains two stages, Acquisition Unsim Frame (AUF) and Acquisition Rare Category (ARC) where AUF aims at acquiring the dissimilar frame, and ARC aims to acquire frame that contains the rare category. We evaluate our model based on three domain adaptation processes, and the result surpasses the current state-of-the-art (SOTA) methods. [1]

## 2   Related Work

*Unsupervised Domain Adaptation for Object Detection.* Unsupervised Domain Adaptation (UDA) [11, 30] aims to mitigate the domain-shift problem without requiring labeled target data, by aligning source and target distributions. The field has seen significant contributions specifically in object detection [1, 15, 24, 26, 31, 32]. For instance, Chen et al. [1] pioneered this space by proposing domain adaptation via both image-level and instance-level alignments. Later, Zhu et al. [31] focused on the alignment of congruent regions between source and target domains. Xu et al. [26] presented a framework that involves categorical

---

[1] The demonstration video is available at:
 https://www.youtube.com/watch?v=3BGsT9iDEGg

regularization at the image level and consistency regularization for categories to address complex challenges in this domain. While these approaches have yielded impressive results, they inherently rely on the availability of source data during adaptation, limiting their applicability in real-world scenarios such as autonomous driving where source data may not always be accessible.

*Source-free Domain Adaptation for Object Detection.* In environments where source data is not readily available, such as in autonomous vehicles, the specialized domain of Source-Free Unsupervised Domain Adaptation (SFDA) has garnered interest [12, 17]. SFDA uniquely operates without the source data during the adaptation phase, and typically focusing on offline model adaptation [5, 7, 10, 21, 22]. The significance of online adaptation in such settings is crucial. Vibashan et al. [23] ventured into this area by proposing a cross-attention transformer-based model for both online and offline SFDA in object detection. They extended the Mean Teacher framework with a MemXformer that pairs features between the teacher and student models [21]. However, despite these advancements, existing methods have not addressed the importance of keyframe selection for adaptation. Our work fills this gap by emphasizing online adaptation in SFDA for object detection, with a particular focus on selective adaptation through keyframe selection. This approach enhances both the efficiency of the adaptation process and the overall adaptation performance.

## 3   Method

### 3.1   Problem Formulation

In the domain of O-SFDA for object detection, we consider an online scenario where access to the original source domain dataset as $D_s = \{x_s, y_s\}$, is restricted, and only a pre-trained model $\Theta_s$ on $D_s$ is available. The aim is to adapt $\Theta_s$ to a target domain $D_t$, which is represented by a stream of unlabeled data $\{x_t\}$. The model is updated using only frames from $D_t$ that are deemed adaptation-worthy.

### 3.2   Base Model

Our method is built upon the framework of online adaptation of the Mean Teacher model [21], specialized for the Faster-RCNN object detector [13]. The Mean Teacher framework employs two models: a student model ($\Theta_S$) and a teacher model ($\Theta_T$), both initialized using the pre-trained source model $\Theta_s$. For each incoming frame $f_t$, we employ two forms of data augmentation: the weak augmentation $\mathcal{A}_w(\cdot)$ and the strong augmentation $\mathcal{A}_s(\cdot)$. The weakly augmented frame $f_t^w$ is fed through the teacher model to generate pseudo-labels, denoted as $\hat{y}_T$, while the strongly augmented frame $f_t^s$ is processed by the student model to predict labels $y_S$. To improve the model's reliability, we consider only pseudo-labels $\hat{y}_T$ that possess a confidence score greater than 0.9, as supported by prior

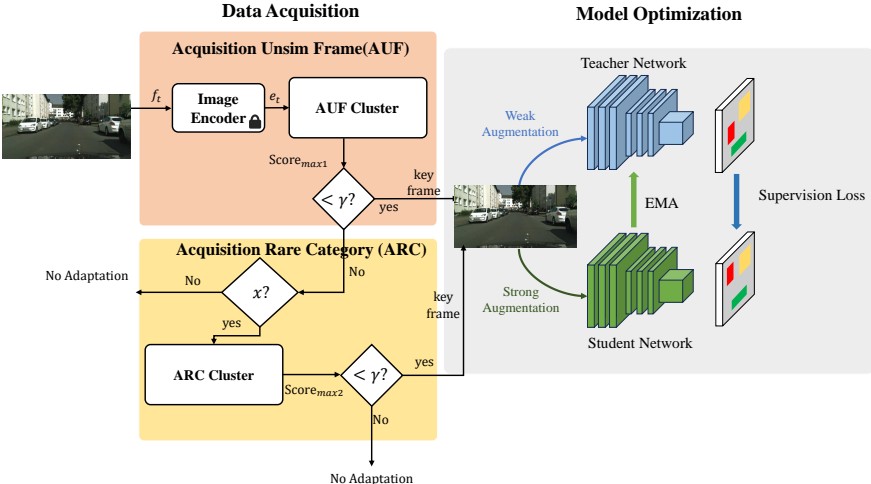

**Fig. 1:** For streaming frames, $f_t$, the data acquisition system continuously evaluates whether to retain the current frame using a similarity measure. Once $f_t$ is detected as "key frame", $f_t$ will be used to update the model. The condition $x$ refers to the if statement used to determine whether the current frame contains the rare category after the warm-up stage.

research [23]. Given this criterion, the loss function within the Faster-RCNN framework can be defined as:

$$L_{\text{FRCNN}} = L_{\text{RPN}}(y_{\text{S}}, \hat{y}_{\text{T}}) + L_{\text{RCNN}}(y_{\text{S}}, \hat{y}_{\text{T}}), \tag{1}$$

### 3.3   Data Acquisition

We present our clustering method to choose frames for adapting the model, especially when data from the source domain is not available. Our data acquisition approach targets two types of data: dissimilar frames and frames containing the rare category. To do this, we divide the process into two stages: the first stage focuses on identifying dissimilar frames by Acquisition Unsim Frame (AUF), and the second stage targets the rare category by Acquisition Rare Category (ARC). Our overall framework is illustrated in Figure 1.

**Acquisition Unsim Frame (AUF)** Leveraging the image encoder $E$ from the static teacher model $\Theta_{\text{T}}$ from the initial teacher parameter, the clustering starts by initializing the first frame $f_0$ as the centroid $p_{11}$ of the first cluster $m_1$. Subsequent frames produce embeddings $e_t$, which are compared against existing centroids $\{p_{11}, \ldots, p_{1n}\}$ using cosine similarity. Based on a predefined similarity threshold $\gamma$, a frame is identified as a keyframe to start a new cluster or is added to the closest existing cluster, $m_c$, which then updates its centroid by $ave(m_c)$. The AUF similarity score, $\text{Score}_{\text{max1}}$ is calculated by:

$$\text{Score}_{\text{max1}} = \max_{i \in \{1, \ldots, n\}} \left( \cos \left( e_t, p_{1i} \right) \right) \tag{2}$$

Frames that meet $\text{Score}_{\text{max1}} < \gamma$ are considered as keyframes for model updates. **Acquisition Rare Category (ARC)** In an online setting, where the model adapts to each current frame, it tends to improve at recognizing frequent objects while performing worse on rarer ones. This issue is known as class imbalance. We addressed this by using the ARC approach, which employs an additional clustering mechanism to give frames with the rare category, $c_r$ a second opportunity for data acquisition. We aggregate the $\hat{y}_{\text{T}\{1,\ldots,t\}}$ and identify the category with the smallest aggregated sum as the rare category:

$$c_r = \arg\min_c (\sum_{i=0}^{t} \hat{y}_{\text{T}i,c} \text{ for each category } c) \qquad (3)$$

Specifically, if $\text{Score}_{\text{max1}} > \gamma$ and $c_r \in \hat{y}_{\text{T}t}$, $e_t$ will be processed into the ARC for a second selection. Similar to the operation in AUF, $e_t$ is compared against existing ARC centroids $\{p_{21}, \ldots, p_{2n}\}$ using cosine similarity:

$$\text{Score}_{\text{max2}} = \max_{i \in \{1,\ldots,n\}} (\cos(e_t, p_{2i})) \qquad (4)$$

Frames meeting the criterion $\text{Score}_{\text{max2}} < \gamma$ are considered as keyframes and trigger model updates. This scheme ensures efficient and effective model adaptation by focusing on frames significantly differ from previous instances. To ensure the stability of the rare category identified, we propose the implementation of a warm-up phase. During the warm-up stage, we only activate AUF and collect the distribution of pseudo labels, $\hat{y}_{\text{T}\{1,\ldots,t\}}$ to identify which category is rare for ARC. This phase is designed to stabilise the rare category prior to activating the ARC mechanism. Notably, the ARC cluster operates independently of the AUF cluster.

### 3.4 Model Optimization

For the optimization of the teacher model and student model, we first use the $\hat{y}_{\text{T}}$ to supervise the student model for parameter update. After the student model is updated, we do an Exponential Moving Average (EMA) to update the teacher model, which transfers the knowledge from the student model to the teacher model. Specifically, to update the student model, we first apply strong data augmentation to produce $f_t^s$ upon identifying a current frame $f_t$ as a keyframe. This augmented frame is then processed by the student model to predict labels $y_{\text{S}}$. Given the absence of true labels in the target domain, we employ pseudo-labels $\hat{y}_{\text{T}}$ generated by the teacher model to adapt the student model parameters $\Theta_{\text{S}}$. A Kullback–Leibler divergence loss [23] $L_{\text{S-T}}$ is also applied to align the $\Theta_S$ classification results, $E_{\text{S}}^{\text{CLS}}$ with the $\Theta_T$ classification results, $E_{\text{T}}^{\text{CLS}}$, based on teacher model's proposal, as follows:

$$L_{\text{S-T}} = L_{\text{KL}}(E_{\text{S}}^{\text{CLS}}, E_{\text{T}}^{\text{CLS}}) \qquad (5)$$

The final loss function $L$ for the student model's update is as follows:

$$L = L_{\text{FRCNN}} + L_{\text{S-T}} \qquad (6)$$

After the student model has been updated by minimizing $L$, we do EMA to update the teacher model. To update the teacher model's weights ($\Theta_T$) using $\Theta_S$, we use two key parameters: $\alpha_1$ for the adaptation phase and $\alpha_2$ for the final update. Specifically, $\alpha_1$ is used in the iterative update during the online process, while $\alpha_2$ is employed for the final model update:

$$\Theta_{T,t} = \alpha_1 \Theta_{T,t-1} + (1 - \alpha_1)\Theta_{S,t-1} \tag{7}$$

$$\Theta_{T,\text{final}} = \alpha_2 \Theta_T + (1 - \alpha_2)\Theta_S \tag{8}$$

## 4    Experiments

### 4.1    Datasets

We evaluate our model using four rigorously selected datasets: Cityscapes [2], Cityscapes-Foggy [16], Sim10k [6], and SHIFT [19].

- **Cityscapes**: Geared towards urban street environments, Cityscapes consists of 2,975 training images and 500 validation images. An additional sequence of 106,917 images from Frankfurt allows for comprehensive evaluations in online sequential adaptation scenarios. We excluded the labelled 267 frames from the Frankfurt sequence to evaluate our model's performance in the target domain. The performance on these 267 frames is reported as "mAP Frankfurt" in the results section.
- **Cityscapes-Foggy**: Cityscapes-Foggy augments the original Cityscapes set with simulated fog, making it an excellent benchmark for evaluating adaptation to adverse weather conditions.
- **Sim10k**: Originating from the video game Grand Theft Auto V, this dataset comprises 10,000 training frames and tests our model's ability to adapt to substantially different source domains.
- **SHIFT**: Created in a simulator, SHIFT encompasses a variety of weather conditions, providing an ideal landscape for assessing the robustness of our model across diverse environments.

**Table 1:** Sim10k to Sequential Cityscapes (Car only)

| Method | Car |
|---|---|
| Source-Only | 31.9 |
| Tent [25] | 33.1±0.3 |
| MemCLR [23] | 37.0±0.2 |
| **Ours** | **46.0±0.2** |

**Table 2:** Shift to Sequential Cityscapes on Frankfurt Validation Set

| Method | Person | Car | Mcycle | Bike | mAP Frankfurt |
|---|---|---|---|---|---|
| Source-Only | 16.1 | 17.1 | 4.5 | 6.1 | 10.9 |
| Tent [25] | 16.5±0.3 | 18.0±0.0 | 8.1±2.2 | 6.5±2.4 | 12.3±1.1 |
| MemCLR [23] | 22.1±0.5 | 25.3±0.5 | 9.9±1.2 | 8.7±0.8 | 16.5±0.4 |
| **Ours** | **30.2±0.8** | **39.5±0.9** | **21.9±3.4** | **11.7±1.2** | **25.8±0.8** |

**Table 3:** Cityscapes to Foggy Cityscapes

| Method | Person | Rider | Car | Truck | Bus | Train | Mcycle | Bike | mAP |
|---|---|---|---|---|---|---|---|---|---|
| Source-Only | 29.3 | 34.1 | 35.8 | 15.4 | 26.0 | 9.09 | 22.4 | 29.7 | 25.2 |
| Tent [25] | 31.2 | 38.6 | 37.1 | 20.2 | 23.4 | 10.1 | 21.7 | 33.4 | 26.8 |
| MemCLR [23] | 32.1 | 41.4 | 43.5 | **21.4** | **33.1** | 11.5 | **25.5** | 32.9 | 29.8 |
| **Ours** | **35.2±0.7** | **44.3±1.3** | **50.5±0.4** | 18.5±3.1 | 30.7±3.7 | **16.1±4.3** | 23.9±1.9 | **37.6±0.3** | **32.1±1.3** |

## 4.2 Implementation Details

Our implementation builds upon the Faster-RCNN framework [13] with a ResNet50 backbone [4], following the baseline established by MemCLR [23]. The ResNet50 model is pre-trained on ImageNet [9]. We employ a batch size of 1 and run the experiment for 1 epoch to ensure online adaptation capabilities. The student model relies heavily on the teacher's predictions for adaptation. To guarantee robustness and keep the same setting with Baseline [23], we only consider teacher-generated pseudo labels with a confidence score exceeding 0.9 as Ground Truth for guiding the student model. The Exponential Moving Average (EMA) parameter is set to 0.996 for $\alpha_1$ and 0.9 for $\alpha_2$. The learning rate is set to 0.001, and the warm-up learning rate is set to 0.0001. The optimization is performed using the SGD optimizer. Our method for acquiring data involves extracting features from the Fix-Teacher model, using a similarity threshold, $\gamma$ set at 0.975 for both AUF and ARC. The warm-up period was adjusted to suit different environments. To build a category occurrence distribution, we run the warm-up until the ratio of the rarest category to the most popular reaches 0.3% when the total pseudo label counts more than 10000.

## 4.3 Evaluation Scenarios

**Online Sequential Adaptation:** Our primary objective is an online adaptation for autonomous vehicles navigating through sequences of overlapping frames. We employ datasets SHIFT, Cityscapes and Sim10k to assess this feature. Detailed evaluation setups include: *(1) Sim10k to Seq-Cityscapes with car category only:* Utilizing Sim10k as the source domain, we evaluate our model's adaptability using the sequential Frankfurt set of the Cityscapes dataset. Since there is only one category, we only activate AUF. *(2) SHIFT to Seq-Cityscapes with shared categories:* In this experiment, SHIFT serves as the source domain to evaluate the model on the sequential Cityscapes dataset. The shared categories for evaluation are person, car, motorcycle, and bicycle.

**Weather Adaptation:** This scenario scrutinizes the model's resilience to sudden environmental changes, particularly shifts in weather conditions. Cityscapes and Cityscapes-Foggy serve as key evaluation grounds. In addition, we introduce an extra evaluation set from the Cityscapes validation set to assess the model's robustness in unseen urban settings. Another reason for this experiment is that Memclr did not mention the sequential scenario. This table aims

**Table 4:** Ablation of Contribution of Each Component

| Method | Person | Car | Mcycle | Bike | mAP Frankfurt |
|---|---|---|---|---|---|
| No acquire | 0.0±0.0 | 5.0±6.8 | 0.0±0.0 | 0.0±0.0 | 1.3±1.7 |
| AUF | 29.5±1.0 | 36.0±5.3 | 16.4±4.0 | 9.0±1.6 | 22.7±2.4 |
| AUF+ARC | **30.2±0.8** | **39.5±0.9** | **21.9±3.4** | **11.7±1.2** | **25.8±0.8** |

for a fair comparison with the existing dataset in MemCLR. We use these results to demonstrate that data acquisition is also effective in weather adaptation. Weather adaptation is non-sequential and involves only 500 frames, making it challenging to stabilize the rare category initially for ARC. Thus, we only consider using AUF.

### 4.4   Comparison with SOTA

We evaluate our model's performance with two SOTA methods **Tent [25]** and **MemCLR [23]** in three specific settings: two types of Online Sequential Adaptation and one Weather Adaptation. Notably, Tent employs entropy minimization to boost model performance in classification and semantic segmentation tasks. We deploy the entropy minimization in object detection by disambiguating the $\Theta_S$ classification result based on the teacher model's proposal. For each experiment, we used random seeds run five times, calculating the mean with standard deviation. We use mAP calculated via the AP50 as the evaluation metric.

As shown in Table 1, when testing the model that was trained on the source data directly, the average precision (AP) for car detection was 31.9, which was the lowest score, as expected. Our model demonstrates a marked improvement over the state-of-the-art method, MemCLR [23], in car detection, achieving an AP score of 46.0±0.2. This is achieved using 345 frames for adaptation, accounting for only about 0.32% of the total available frames. As illustrated in Table 2, we focus on four share categories: person, car, motorcycle, and bicycle, reporting results in terms of mAP. Our model surpasses MemCLR in performance by 9.3 mAP while utilizing only approximately 384 frames. Significantly, our method greatly improved the detection of motorcycles, a rare category, by achieving 12 AP higher than MemCLR [23]. As shown in Table 3, our model outperforms MemCLR with an mAP score of 32.1, improving by 2.3 points. Notably, the AP score for the car category increased by 7 points compared to MemCLR's performance.

### 4.5   Ablation Study

Our study places significant emphasis on sequential datasets, particularly the transition from the SHIFT to the Cityscapes dataset, to simulate real-world conditions more effectively. This section elucidates the effects of various experimental components on the final performance metrics of our model.

Table 4 illustrates the effects of various configurations on the overall model performance. The term "No acquire" refers to our method when using all the frames for adaptation. Resulting in a severe collapse of the performance, likely due to frame overlaps and redundant information. The configuration labelled "AUF" involves the use of dissimilar frame acquisition alone and achieved a mean Average Precision (mAP) of 22.7, which is an improvement of 21.4 mAP over the "No acquire" setup. The "AUF+ARC" configuration, which incorporates both dissimilar frame acquisition and rare category acquisition, demonstrated superior performance compared to "AUF". Specifically, it showed notable improvement in the rare category, with motorcycles achieving an AP of 5.5. Moreover, the overall mAP increased by 3.1 compared to the "AUF" configuration.

### 4.6  Efficiency and Time Cost

Data acquisition not only enhances model performance in the target domain but also reduces the time needed for adaptation. We evaluated the performance speed of MemCLR and our method on the sequential Cityscapes dataset, measuring the average processing time per frame in milliseconds (ms). Our method demonstrated an efficiency of 12.7 ms per frame, compared to MemCLR's 21.3 ms, with both methods tested on RTX 4090 GPU. This is because MemCLR requires both inference and adaptation for each frame, while our method only performs inference and selects key frames, avoiding constant adaptation.

## 5  Conclusion

In this work, we introduce an unsupervised data acquisition method in O-SFDA object detection tailored to autonomous vehicles in diverse and dynamic settings. This approach operates in two stages: initially, it selects highly informative frames, and then it gives extra attention to those frames that also include the rare category to reduce class imbalance and frame overlap. Experimental results on multiple datasets show the effectiveness of our method and outperform existing methods in terms of object detection performance.

## Acknowledgement

We thank Yang Zhao for his assistance in video creation.

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

# A   Appendix

## A.1   Qualitative study

Figure 2 displays the visualization results from the Cityscapes validation set for Frankfurt city, showing bounding boxes with confidence scores above 0.5. We compare Source-only, Tent, and MemCLR with our method. Bounding boxes are colour-coded: red for the person, green for the car, blue for the motorcycle, and yellow for the bike. The comparison of these four images clearly demonstrates the superior performance of our model.

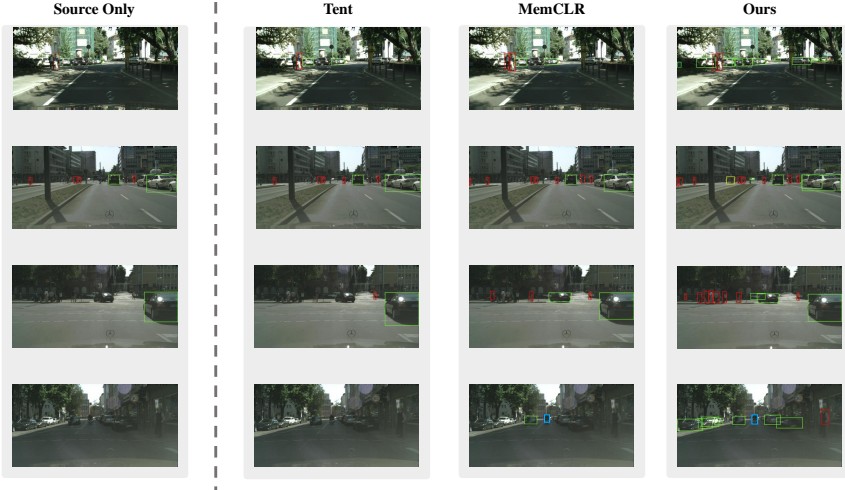

**Fig. 2:** The four qualitative results demonstrate the performance of each baseline and our method on the Cityscapes validation set. The colours of the bounding boxes indicate different objects: red for **Person** , green for **Car**, blue for **Mcycle**, and yellow for **Bike**.