# OpenReview forum: "Improving Online Source-Free Domain Adaptation for Object Detection by Unsupervised Data Acquisition"
_thecvf.com/ECCV/2024/Workshop/ROAM — ROAM ECCV 2024 Oral_

### Official Review · Reviewer_ruSY · 2024-08-08

**Rating:** 6
**Confidence:** 3

**Review:**

The paper proposes a domain adaptation method that dynamically selects frames on-the-fly to improve the performance of a network trained on an inaccessible source domain. I found the overall pipeline to be reasonable, and it appears that the network's performance has improved on the test domain.

However, I believe the paper would benefit from a clearer definition of some important processes, such as what constitutes weak and strong augmentation, as well as a detailed explanation of the losses defined in Eq. (1). Providing these clarifications would make the model more understandable and the paper more self-contained.

---

### Official Review · Reviewer_qwRL · 2024-08-10
**Review of submission 3**

**Rating:** 6
**Confidence:** 4

**Review:**

The paper addresses the challenge of Online Source-Free Domain Adaptation for object detection in autonomous vehicles. The authors propose a novel two-stage unsupervised data acquisition method to enhance adaptive object detection. The method involves selecting the most informative unlabeled frames, with a focus on rare categories, to mitigate class imbalance and reduce computational costs. Empirical evaluations across multiple datasets demonstrate the performance.

Thank you for submitting your paper. I hope my comments are helpful and contribute to further enhancing your work.

# Following my comments:
The dynamic selection of informative frames is an interesting approach. This is practical, especially in scenarios where computational resources are limited, and class imbalance is a concern. The paper is also well structured, and the results are comprehensive.

Despite the strengths, the paper has some weaknesses in the presentation of the method. First, the concept of "informative frames" is inadequately explained, leaving readers unclear on what constitutes an informative frame. The explanation of keyframes and the criteria for selecting them, particularly in relation to handling domain shift, lacks clarity. A literature review could have been beneficial, highlighting the differences with existing works.

The empirical results, although promising, do not fully explore the generalizability of the method across different scenarios or environments, leaving gaps in understanding the broader applicability of the approach. For example, would have been interesting to show results in the dataset with higher domain gap.

The paper mentions the challenge of adapting to new domains "on-the-fly" but does not really discuss about it.

Clarifications are also necessary regarding the results in Tables 1-3: it should be specified whether these are based on the authors' implementations of Tent and MemCLR or if they are results directly cited from existing literature. Additionally, the significant difference in gains observed in Table 2 as compared to other datasets needs explanations to help in understanding these discrepancies. why the gains are much more different than from the other results?

---

### Official Review · Reviewer_X9rd · 2024-08-12
**Review of submission 3**

**Rating:** 6
**Confidence:** 4

**Review:**

The authors propose an online source-free domain adaptation methodology designed to enhance the training process of object detection models through unsupervised learning by discriminating between key frames. The goal of developing strategies to adapt models to new domains without relying on labeled data is both relevant and clearly motivated, and the proposed approach appears to be novel and interesting. However, I have a few concerns that need to be addressed:

1. Terms such as “weak adaptation” and “strong adaptation” need a more thorough explanation. Additionally, the concept of “rare category” should be defined more precisely. Specifically, it should be clarified whether “rare category” refers to pre-existing categories with low detection scores or to new, additional categories in the dataset.

2. An analysis of the computational complexity related to the number of centroids as the duration of the video increases would provide insight into the efficiency and scalability of the proposed method.

3. Authors could elaborate on how this methodology relates to existing literature on anomaly detection. Understanding the connection could provide valuable context for the proposed approach.

4. It would be helpful to discuss how the percentage of key frames is adapted based on desired latency and available computational resources.

Addressing these points will enhance the clarity and comprehensiveness of the proposed methodology.

---

### Official Review · Reviewer_XSKE · 2024-08-13
**Review of submission 3**

**Rating:** 5
**Confidence:** 3

**Review:**

This paper explore Online Source-Free Domain Adaptation for object detection. The main contribution is that an unsupervised data acquisition method is designed. The data acquisition process includes two metrics, AUF and ARC. These two metrics are shown effective in the experiments.

The paper is hard to follow and many details in model and experiment setting are missing. Why is the result of "No acquire" setup near to 0? The authors explain that it is "due to frame overlaps and redundant information". However, it cannot convince me. Besides, no ablation study for hyper-parameter is conducted, which is importance for such a heuristic method.

---

### Official Review · Reviewer_HK8r · 2024-08-16
**Review of submission 3**

**Rating:** 5
**Confidence:** 4

**Review:**

This paper proposes an online source-free domain adaptation method for object detection in the context of autonomous vehicles. Overall, the main idea consists of integrating two ‘decision’ branches to find ‘key frames’ for updating models within a teacher/student framework.

Pros:

The idea is interesting (however I’m  not certain about its novelty) and it seems relevant for the context of autonomous vehicles.

The proposed method has been evaluated with multiple datasets, which contributes towards its applicability.

Cons:

I strongly believe that this paper needs to include a related work section. It is not clear for me how their proposal differs from existing works. What is the real contribution here? Some parts of the system seem very similar to [17]. Where are the ideas from ARC and AUF clusters coming from? Are ARC and AUF something that all readers should know about? Or are these components firstly introduced in this paper? If yes, more details about these would be needed.

I have some concerns regarding the fairness of evaluation. It does not seem right to me. Why should one expect inferior performance if the model is strategically selecting frames with mostly correct detections? Why has only one ‘rare’ class been included? How is standard evaluation performed on this setting?

Would be good to include arrows on table headers to better understand performance comparisons. What does the symbol on the Tent method mean? Captions for all tables could be more descriptive.

I believe ablation studies should be extended. Since it seems that ARC and AUf are the main components of the proposal. What would happen when including those to other methods, e.g. [17]?

Why is the reason to set gamma to 0.975? This leads to another concern, why there are not experiments with different hyperparameters?

Overall, the paper reads well, but there are some parts that could be improved. For example:

Line 006: Not sure about this statement. Why deploying object detection methods in unfamiliar environments is an issue? I think the issue here is not with the deployment itself, if not with the behaviour/performance of models under environments that differ a lot from the data distribution of training.

Line 021: First 2 sentences of this paragraph do not follow. I don’t see what is the logical relationship between them. I’d recommend rewriting it.

Line 043: Why are only motorbikes mentioned as ‘rare classes’? I believe there are many other examples.

Line 054: Here it is mentioned that the model will be evaluated with a ‘three domain adaptation process’. Later in line 067, these are referred to as benchmarks. It would be beneficial for readers to have some kind of consistency for this terminology.

Caption of figure 1 is unclear. Please have a look at the part that describes x, it is hard to understand what this means.

---

### Decision · Program_Chairs · 2024-08-22

**Decision:**

Accept (Oral)

**Comment:**

The average score of the paper given all the reviews received before the deadline was higher than 5.5 (1 is lowest, 10 is highest), therefore the paper is accepted. The Authors are encouraged to consider feedback for the camera ready version of the paper due on August 31st.